Citation analysis of computer systems papers

Frachtenberg Eitan etc_26@yahoo.com
Computer Science, Reed College , Portland , OR , United States of America
Galán José Manuel
Electronic publication date: 2023 May 16
Publication date: 2023
Volume: 9
Electronic Location ID: e1389
Received 2023 Mar 9; Accepted 2023 Apr 20
Copyright: ©2023 Frachtenberg
Copyright year: 2023
Copyright holder: Frachtenberg
License: This is an open access article distributed under the terms of the Creative Commons Attribution License, which permits unrestricted use, distribution, reproduction and adaptation in any medium and for any purpose provided that it is properly attributed. For attribution, the original author(s), title, publication source (PeerJ Computer Science) and either DOI or URL of the article must be cited.
License URL: https://creativecommons.org/licenses/by/4.0/

Keywords: Citation analysis, Computer systems, Digital libraries, Self-citations

Funding: The author received no funding for this work.

==============================
Citation analysis is used extensively in the bibliometrics literature to assess the impact of individual works, researchers, institutions, and even entire fields of study. In this article, we analyze citations in one large and influential field within computer science, namely computer systems. Using citation data from a cross-sectional sample of 2,088 papers in 50 systems conferences from 2017, we examine four research areas of investigation: overall distribution of systems citations; their evolution over time; the differences between databases (Google Scholar and Scopus), and; the characteristics of self-citations in the field. On citation distribution, we find that overall, systems papers were well cited, with the most cited subfields and conference areas within systems being security, databases, and computer architecture. Only 1.5% of papers remain uncited after five years, while 12.8% accrued at least 100 citations. For the second area, we find that most papers achieved their first citation within a year from publication, and the median citation count continued to grow at an almost linear rate over five years, with only a few papers peaking before that. We also find that early citations could be linked to papers with a freely available preprint, or may be primarily composed of self-citations. For the third area, it appears that the choice of citation database makes little difference in relative citation comparisons, despite marked differences in absolute counts. On the fourth area, we find that the ratio of self-citations to total citations starts relatively high for most papers but appears to stabilize by 12–18 months, at which point highly cited papers revert to predominately external citations. Past self-citation count (taken from each paper’s reference list) appears to bear little if any relationship with the future self-citation count of each paper. The primary practical implication of these results is that the impact of systems papers, as measured in citations, tends to be high relative to comparable studies of other fields and that it takes at least five years to stabilize. A secondary implication is that at least for this field, Google Scholar appears to be a reliable source of citation data for relative comparisons.

Introduction

Citation analysis plays a central role in bibliometric evaluation of journals, conferences, institutes, and individual researchers (Moed, 2006). The advent of web-based citation databases has led to faster growth in their importance and use (Meho, 2007). Nonetheless, citation analysis remains challenging when comparing citations across years (Varga, 2019), types of scholarly communication (Martins et al., 2010), or fields of study (Adam, 2002; Patience et al., 2017).

This study explores a multifaceted citation analysis that circumvents these challenges by focusing on papers from a single year, from conferences only, and from a single large field within computer science (CS), namely, computer systems (or just “systems” for short). Citation analysis can play an increasingly important role in judging and quantifying the importance of scientists and scientific research (Meho, 2007), as well as in policy making (Moed, 2006). Concomitantly, the debate about citations’ utility and interpretation appears to increase just as much (MacRoberts & MacRoberts, 1989; MacRoberts & MacRoberts, 2018). In computer science and computer systems, citation analysis often focuses on conference proceedings in particular, since they are seen as more timely, more cutting-edge, and more strictly refereed than some journals (Goodrum et al., 2001).

The bibliometrics literature is rich with studies analyzing citations in various disciplines and fields, including CS as a whole (Devarakonda et al., 2020; Hirst & Talent, 1977; Mattauch et al., 2020). Even within CS, several fields, subfields, and specific venues have received bibliometric analyses (Broch, 2001; Frachtenberg, 2022a; Iqbal et al., 2019a; Iqbal et al., 2019b; Lister & Box, 2008; Rahm & Thor, 2005; Wang et al., 2016). The purpose of this article is to apply these methods to the CS field of computer systems and to understand the distribution and characteristics of its citations. To the best of the author’s knowledge, this study is the first systematic analysis of a wide cross-section of computer systems research.

Systems is a large research field with numerous applications, used by some of the largest technology companies in the world. For the purpose of this study, we can define systems as the study and engineering of concrete computing systems, which includes research topics such as operating systems, computer architectures, data storage and management, compilers, parallel and distributed computing, and computer networks. The goal of this study is to characterize, for the first time, the citation behavior of this large and important CS field, while controlling for year, venue type, and research area. Specifically, this article examines: total citations after five years and how they compare across subfields and against other disciplines and fields; the dynamics of citation counts over time; the effect of the source of the citation database; and the characteristics of self-citations in computer systems.

This study uses an observational, cross-sectional approach, analyzing 2,088 papers from a large subset of leading systems conferences. The study population came from a hand-curated collection of 50 peer-reviewed systems conferences from a single publication year (2017). Among other characteristics, the dataset includes paper citation counts at regular intervals for at least five years from the publication date, by which time many citation statistics stabilize (Larivière, Gingras & Archambault, 2009). More details on the collection methodology and selection criteria can be found in previous work (Frachtenberg & Kaner, 2022). Using this dataset, this study addresses the following high-level research questions:

RQ1: What is the distribution of paper citations in systems conferences after five years?

Citations in this dataset exhibit the typical skewed distributions with most papers accruing a few citations, a handful of papers racking up thousands of citations, and a dearth of uncited papers. According to one comparison, citation counts of top-cited papers put the field of systems among the top ten scientific fields. However, citations are distributed unevenly across and within conferences and subfields, with security, databases, and computer architecture conferences ranking the highest median citations.

RQ2: How have citations evolved over this period?

In this dataset, the median number of citations per paper grows at a nearly constant rate, with few papers peaking before the five-year mark, and few papers achieving “runaway” citation growth. Most papers are first cited within 9–12 months since publications, and the average time to have to wait to reach at least n citations grows almost linearly with n.

RQ3: How do Google Scholar and Scopus compare for these conferences?

The data suggests, as have other studies before, that the Google’s inclusive policy of counting many document types as potential citation sources does inflate the absolute citation counts compared to Scopus’ counts. However, the two counts are nearly perfectly correlated and provide similar relative comparisons across papers, conferences, and years.

RQ4: How many citations are self-citations, and how do they evolve over time?

The main finding emerging from the dataset is that self-citations are more prevalent among early citations and among papers that are less cited overall after five years. However, the average ratio of self-citations to total citations appears to stabilize for most papers about 12–18 months after publication. This dataset shows no clear relationship between a paper’s eventual self-citations and the number of self-citations it contains in its own reference list.

In addition to answering these research questions, this study makes the following contributions:

• A critical view of H-index as a metric for conferences or journals.

• A discussion of the relationship between acceptance rates and citations.

• A characterization of papers that are cited relatively early.

As a final contribution, this study provides the dataset of papers and citations over time, tagged with rich metadata from multiple sources (Frachtenberg, 2021). Since comprehensive data on papers and conferences with citations are not always readily available, owing to the significant manual data collection involved, this dataset can serve as the basis of additional studies (Saier, Krause & Färber, 2023).

The remainder of this article is organized as follows. The next section ‘Materials and Methods’ describes the data collection and processing methodology in detail. The results section ‘Results’ enumerates the findings, organized by research question. ‘Discussion’ combines results on raw citation statistics to explore three higher-level topics. Related work is surveyed in ‘Related work’, and ‘Conclusion and future work’ summarizes the results and suggests directions for future research.

Materials and Methods

The most time-consuming aspect of this study was the collection and cleaning of data. This section describes the data selection and cleaning process for conference, paper, and citation data.

Data were collected as previously described in Frachtenberg (2022b). Specifically, the dataset comes from a hand-curated collection of 50 peer-reviewed systems conferences from a single publication year (2017) to reduce time-related variance. Conference papers were preferred over journal articles because in CS, and in particular, in its more applied fields, such as systems, original scientific results are typically first published in peer-reviewed conferences (Patterson, Snyder & Ullman, 1999; Patterson, 2004), and then possibly in archival journals, sometimes years later (Vrettas & Sanderson, 2015). These conferences (detailed in Appendix A) were selected to represent a large cross-section of the field with different sizes, competitiveness, and subfields. Such choices are necessarily subjective, based on the author’s experience in the field. But they are aspirationally both spread enough to represent the field well and focused enough to distinguish it from the rest of CS. For each conference, various statistics were gathered from its web page, proceedings, or directly from its program chairs.

Since the main interest of this article is in measuring the posthoc impact of each paper, as approximated by its number of citations, citation data was regularly collected from two databases, Google Scholar (GS) and Scopus. GS is an extensive database with excellent coverage of CS conferences that contains not only peer-reviewed papers, but also preprints, patents, technical reports, and other sources of unverified quality (Halevi, Moed & Bar-Ilan, 2017). Consequently, its citation counts tend to be higher than those of databases such as Scopus and Web of Science, but not necessarily inferior when used in paper-to-paper comparisons (Harzing & Alakangas, 2016; Martin-Martin et al., 2018). Since we are mostly comparing relative citation metrics, even if the GS metrics appear inflated compared to other databases, we should still be able to examine the relationship between relative citation counts of papers and conferences. Nevertheless, for papers covered in the Scopus database, both sources of citations are compared in RQ3 to ensure that both metrics are in relative agreement with each other.

To collect citation statistics from GS, each paper’s citation count was collected from https://scholar.google.com/ at each monthly anniversary during the first year since publication. During the second year, the statistics for each paper were collected every 3 months, and afterwards, every six months. Scopus, on the other hand, offers retroactive citation statistics, so its citation and self-citation data were collected for the end of each calendar year, as well as in each paper’s five-year anniversary.

Statistics

For hypothesis testing, group means were compared pairwise using Welch’s two-sample t-test and group medians using the Wilcoxon signed-rank test; differences between distributions of two categorical variables were tested with the χ2 test; and correlations between two numerical variables were evaluated using Pearson’s product-moment correlation coefficient. All statistical tests are reported with their p-values. All computations were performed using the R programming language and can be found in the source code accompanying this article.

Ethics statement

All data for this study was collected from public online sources and therefore did not require the informed consent of the authors. No funding was received to assist with the preparation of this manuscript.

Code and data availability

The complete dataset and metadata are available online (Frachtenberg, 2021). For ease of reproducibility, a Docker image with the source code and dataset is also available at https://hub.docker.com/r/eitanf/sysconf with the tag ‘citations’.

Limitations

To control for the effect of time on citations, additional data from more recent conference years in the dataset was excluded. Undoubtedly, more data could strengthen the statistical validity of the observations; but it could also weaken any conclusions based on the inherent delays in the citation process and in variation over time. The methodology is also constrained by the manual collection of data, such as conference statistics; paper downloading and text conversion; cleanup and verification; etc. The effort involved in compiling all necessary data limits the scalability of this article’s approach to additional conferences or years.

The focus on citations as a primary metric of interest has also received significant criticism in the bibliometrics literature because of its own limitations. First, there are limitations stemming from the sources of citation data, such as inflated metrics in GS and partial coverage in Scopus, as previously mentioned. Second, even accurate citation counts present a variety of limitations, such as references expressing negative sentiments (Parthasarathy & Tomar, 2014) or researchers and venues gaming their own citation counts (Biagioli, 2016). Nevertheless, citations are arguably the most popular method to evaluate research impact, either as raw counts or as inputs to compound metrics, such as the H-index. As such, this initial analysis of citation behavior in a previously unexplored and important field could further our understanding of both the particular field and bibliometrics as a whole.

Results

RQ1: Citation distribution

Let us begin by describing the distribution of citations in the field. In addition to observing summary statistics, such as means, medians, and outliers, focusing our attention on distributions can increase transparency when comparing citation metrics because of their typical skewness (Larivière et al., 2016). To better understand the distribution of 5-year citations, starting top-down by looking at the overall distribution and characteristics of all cited papers and then zoom in on distributions by conference and by paper topics. The missing piece is then filled by describing the distribution of uncited papers.

Overall distribution

Figure 1 shows the overall distribution of cited papers exactly five years from their publication date. It is indeed skewed and long-tailed (note the logarithmic scale) and resembles a log-normal distribution (Rahm & Thor, 2005; Redner, 1998; Wang & Barabási, 2021; Wu, Luesukprasert & Lee, 2009). Consequently, the mean number of citations, 55, is much higher than the median, 23, and even higher than the 75th percentile, 53. The mean is pushed this high by a handful of outlier papers: there are just seven papers with over 1,000 citations each, one as high as 5229! Similarly, we find as many as 267 papers (or 12.8% of the dataset) with at least 100 citations. This large group of papers would probably no longer be considered an outlier, and 100 citations are likely a reliable signal of notable impact, again suggesting that the field as a whole may be quite influential.

If we follow Patience’s method (Patience et al., 2017) to attenuate the outliers and compute the mean number of five-year citations among the top-cited 31–500 papers, we get an average of 120. That same study also compared this statistic across 236 science fields. So, using their data for comparison would rank systems as one of the top ten fields in citation count (but keep in mind that the original comparison used less-inflated citations counts from the Web of Science database.)

Citations by conference

Although overall citation counts for computer systems appear high, keep in mind that not all conferences within the field are equally well cited. Figure 2 shows the same kind of histogram, but this time broken down by conference and sorted by median citation per conference. We can observe the following relationships from the data:

Figure 1 Citation distribution after five years of all cited papers.

Also shown are the number of samples (papers), mean, and median citations per paper.

Figure 2 Citation distribution of all cited papers by conference.

Also shown are the number of samples (papers), mean, and median citations per paper. Conferences ordered by median and showing acceptance rate in parenthesis next to their names.

• Citations vary widely between conferences, ranging from an average of about four for HCW papers to about 152 for SIGCOMM papers.

• Even within conferences, some outlier papers push the mean far from the median. For example, SP and ISCA exhibit long tails, with a few papers having thousands of citations, whereas NDSS, an equally well-cited conference, exhibits nearly identical mean and median.

• The likelihood of a conference containing an outlier paper with at least 500 citations increases with conference size. Papers with over 500 citations were published in conferences averaging 85.7 accepted papers, vs. an average of 58.6 for papers with fewer than 500 citations (t = 2.47), (p = 0.02). Furthermore, a conference’s size is weakly but positively correlated with its average paper citations (r = 0.32), (p = 0.03), suggesting a random variable in the citation count of a paper.

• Median citations, on the other hand, do not appear to be significantly correlated with a conference’s number of accepted papers (r = 0.24), (p = 0.09). However, conferences with higher median citations are generally more competitive (i.e., lower acceptance rates). These two factors exhibit a strong negative correlation (r = −0.64), (p < 10−5). Likewise, mean citations per conference is also negatively correlated with the conference’s acceptance rate (r = −0.59), (p < 10−5).

• Different conferences have different skews: the distribution can be wide or narrow; some have long right tails and some have none; and the modes appear at different locations, not always near the median. In other words, despite all conferences belonging to the same overall systems field, citation distributions still vary significantly with other factors including size, acceptance rate, and specific subfield.

• Related, some conferences sharing subfields of systems appear to be also clustered together in citation metrics. For example, NDSS, CCS, and SP—all focused on computer security—are similarly ranked in terms of median citations (although the means vary much more). Likewise, NSDI/SIGCOMM (focusing on networking), HPCA/MICRO and ASPLOS/ISCA (focusing on computer architecture) also pair with similar median citations.

This last observation naturally leads to the question of the relationship between a research subfield and its typical citations. To answer this question, let us next look at research topics at the individual paper level rather than at the conference’s broad scope.

Citations by subfield

The definition of subfields in computer systems, like the definition of the field itself, is necessarily fluid and subjective. Based on research experience in the field, a best-effort attempt was made to categorize all papers by reading every single abstract and tagging each paper with one or more subfields. The list of selected tags is presented in Table 1. Naturally, experts would differ in their opinions on this list and possibly on tag assignments to papers, but the current assignment provides a starting point for comparison across systems subfields.

Table 1 List of selects systems subfields.

Tag	Subfield description	
Architecture	Computer architecture	
Benchmark	Workloads, bechnmarking, metrics, and methodology	
Cloud	Cloud computing and related infrastructure	
Compilers	Compilers, language and debugging sopport, runtime systems	
Concurrency	Parallel and distributed computing	
Data	Big data applications and infrastructure	
DB	Databases, key-value stores, and database management systems	
Energy	Power and energy efficiency, sustainable computing	
GPGPU	Accelerator technologies and heterogeneous computing	
HPC	High performance computing and supercomputing applications	
Network	Networking algorithms, wireless networks, switching and routing	
OS	Operating systems, scheduling, resource management	
Security	Security, privacy, encryption, resilience	
Storage	Persistent and ehpemeral storage: disk, flash, RAM, etc.	
VM	Containers and virtual machines and, networks	

Papers in this dataset are generally focused on one or two of these topics, averaging 1.7 topics per paper. There are a total of 2,026 (97% of all papers) cited papers with at least one topic assigned. The remainder either had no citations or did not fit well with any of the tags. Figure 3 shows the distribution of citations by topic in these tagged papers (with multitopic papers repeated).

Figure 3 Citation distribution for all cited papers by paper’s topic.

Papers with multiple topics appear in multiple histograms. Also shown are the number of samples (papers), mean, and median citations per paper. Topics ordered by median citations per paper.

Again, there appears to be a positive relationship between quantity—the number of papers published on a topic—and “quality”—how well cited these papers generally are (r = 0.57), (p = 0.03). This could again indicate a random element leading to a higher probability of outlier papers in large subfields, or it could indicate that these topics were both sufficiently popular at the time to attract multiple submissions and popular over time to attract eventual citations.

Nevertheless, this correlation should also be considered with a grain of salt because of the subjective process of conference selection for this study. For example, if more conferences and workshops on virtualization had been chosen, the number of papers on the topic would have obviously increased, whereas the relatively large conferences on security skew topic counts from the other end. At any rate, in 2017, the broad research topics that resulted in the highest median citations appear to be security, databases and data management, computer architecture, and networks.

Uncited papers

Owing to the logarithmic citation scale, the preceding histograms omitted papers with zero citations. Some early studies claimed that, generally, most scientific papers are not cited at all (Hamilton, 1991; Jacques & Sebire, 2010; Meho, 2007). More recent research found that the rate of uncited papers keeps decreasing, and estimates it to be less than 10% (Wu, Luesukprasert & Lee, 2009). For example, one study computed the percentage of uncited papers in physics (11%), chemistry (8%), and biomedical sciences (4%) (Larivière, Gingras & Archambault, 2009). Another large-scale estimate for the entire CS discipline found that 44.8% of CS papers remained uncited after five years (Chakraborty et al., 2015). In this dataset of systems papers, only 32 papers remain uncited after five years (1.5%). Of these, the conference with the most uncited papers was HPCC, followed by IPDPS and EuroPar (Table 2). Both in absolute terms and in percentage terms, the number of uncited papers remains very low for most conferences.

Table 2 Uncited papers by conference, as paper count and percentage of all accepted papers.

Conference	Count	Percent	
HPCC	6	7.79%	
IPDPS	4	3.45%	
EuroPar	3	6%	
Cluster	2	3.08%	
HiPC	2	4.88%	
ICPE	2	6.9%	
ISPASS	2	8.33%	
CCGrid	1	1.39%	
CLOUD	1	3.45%	
HCW	1	14.29%	
HotStorage	1	4.76%	
HPCA	1	2%	
HPDC	1	5.26%	
ICPP	1	1.67%	
IGSC	1	4.35%	
IISWC	1	3.23%	
PODC	1	2.63%	
SYSTOR	1	6.25%	

The topic tags with the most uncited papers were Benchmark, Storage, and HPC, followed by GPGPU and Concurrency (Table 3). It is not surprising that the two distributions appear to be related. Many papers in the top six uncited conferences were tagged with some of the top six uncited topics.

Table 3 Uncited papers by topic tags.

Topic	Count	
Benchmark	8	
HPC	7	
OS	7	
Storage	7	
Concurrency	6	
GPGPU	6	
Data	5	
Network	4	
Energy	2	
Architecture	1	
Cloud	1	
Compilers	1	

Having examined the citation distribution at a fixed point in time, we can now examine how it evolved to this point from the date of publication.

RQ2: Citation dynamics

Observing the total citations of papers at a fixed time point offers only a static view of a metric that is inherently a moving target. Citations tend to follow different dynamics as different papers, disciplines, and fields exhibit very different aging curves (Pichappan & Ponnudurai, 1999; Wang, 2013). After five years, all the papers in this dataset likely had a chance to be discovered by fellow researchers, as evidenced by the fact that nearly all are cited by outside researchers. We can therefore concentrate next on citation dynamics.

For the second research question, three aspects of the time dimension are examined: general distribution over time, time to first citation, and citation velocity. Since different conferences publish in different months, causing nearly a year’s gap between the first and the last, time is normalized for all three aspects by counting the number of months passed from the date it was published, rather than a fixed start day in 2017.

Citation distribution

Looking at the citation distribution at six-month intervals (Fig. 4), three observations can be made. First, we can see that the spread—difference across papers—grows over time (note the logarithmic scale), as some papers accelerate at a faster rate than others, creating a larger range of citation values.

Figure 4 Citation distribution for all cited papers over time.

Center bars show median value; lower and upper hinges correspond to the 25th and 75th percentile; upper points show high-citation outliers; and diamonds show means for all citations, including uncited papers. Also shown in gray is a linear regression line for the median values.

Most citations (as seen in the first, second, and third quartiles) grow at a rapid clip over time without showing signs of slowing down (note the logarithmic scale). In fact, if we fit the median with a simple linear regression model, shown as a gray line, we match the median citation values after the first year nearly perfectly (intercept: −2.33, slope: 0.43 citations per month). Linear models for the 25th and 75th percentiles would have different slopes, because of the increasing spreads, but still fit quite well.

In contrast, mean citation outpace medians’ growth because of the disproportionate pull of outlier papers. Except for the first six months (when the difference between the mean and the median is less than half a citation), the presence of highly cited papers pushes the mean increasingly higher than the median. In the first six months, the presence of many (1,191) uncited papers is pulling the mean to ≈ 1.6 citations. However, as papers gain enough time to be discovered, cited, peer-reviewed, and then published, the mean quickly catches up. By 12 months, the mean number of uncited papers drops to 503, and by 18 months to 227.

This observation naturally leads us to the next aspect of time: how long does it take papers to be cited?

Time to first citation

The dataset includes citation data at monthly intervals during the first year from publication, then quarterly for the second year, and then every six months for the remaining three years. This resolution allows us to estimate with near-monthly accuracy the first time that GS detected a citation for each paper. Figure 5 shows the distribution of the approximate time in months it took GS to first detect any citations for each paper (compared to Fig. 1, two papers briefly showed citations before reverting back to zero).

Figure 5 Distribution of time to first citation for all cited papers.

Also shown are the number of samples (papers), mean, and median times in months.

Assuming that the first citation is external, i.e., not a self-citation, we can think of this moment of first being cited as a paper’s “discovery event.” In this dataset, it averages about 9 months. (As we see later in RQ4, this assumption is invalid for many papers, which may postpone the discovery time by a few months.) Discovery time can be longer than subsequent citations because it requires the paper to be published and discovered by another researcher, who must then wait till their citing document is peer-reviewed and published, a process that can take several months.

To illustrate this point, Fig. 6 shows the mean time it takes for papers to reach n citations. Note that the average growth from the first citation to the next and then the next takes about the same time across the range (about 2–3 months for each additional citation, slowly decreasing). However, going from zero citations to the first citation takes on average about twice as long. This observation leads us to the next and final question of time, namely, how fast and how long do citations grow?

Figure 6 Average time to reach n citations for increasing values of n.

Only papers with at least 10 citations after five years are included.

Citation velocity

As we have just observed, the mean time across papers to receive the first citation (6.24 months) is longer than the time to add the second citation (2.91 months). It continues to decrease slowly such that it only takes 1.86 months on average to add the 10th citation. Intuitively, this makes sense as papers are discovered by an increasingly larger network of researchers and pick up more and more citations during their growth phase. However, this phase cannot grow indefinitely, as the size of the potential network is bounded by the systems research community size, and the impact of systems papers is often limited to a few years until newer systems replace them. To visualize these growth patterns, we can plot the citation velocity of every paper (Fig. 7).

Figure 7 Citation velocity of every paper.

Y-axis is clipped to show the bulk of the data with higher resolution, but some papers exceed 100 citations per month. Conferences ordered again by increasing median citations after five years.

The first observation we can make from the figure is that the citation growth for most papers is nominal, as evidenced by the thick near-flat bands of most papers at every conference. The growth rate hovers close to zero for conferences with low median citations and remains mostly below five citations per month even for better-cited conferences. The second observation is that a few papers do achieve runaway growth, sometimes exceeding the plot limit of 15 citations per month—even reaching 147 in one instance—but their overall number is small. Only 26 papers exceed ten citations per month at the end of the five years.

This brings us to the last observation and provides an answer to the first question posed. There were actually slightly more papers exceeding ten citations per month at the four-year mark (30 papers), meaning that a few of those peaked before reaching five years. Looking at the graphs (ignoring some sharp peaks and dips caused by noise from GS), we can observe the occasional paper that reached maximum velocity and then started to slow down. Again, the overall number of these appears small, suggesting that most papers in systems take longer than five years to peak.

RQ3: Comparison of GS and Scopus citations

The citation criteria employed by GS tend to be more inclusive than those of other databases, sometimes resulting in inflated citation counts (Harzing & Alakangas, 2016; Martin-Martin et al., 2018). Moreover, different fields tend to be covered to different degrees by different databases, and not all databases cover conferences equally well (the systems conferences in this dataset are covered 100% by GS). Because of this criticism of GS and as a way to add a measure of control to the findings, we can examine citation data from another database, Springer’s Scopus.

Two types of statistics were collected from Scopus: total citations exactly five years from each paper’s publication and total citations at the end of each year (with and without self-citations). We will first focus on the former statistic, which is directly comparable to the one collected from GS, and explore the latter in the next research question.

Note that Scopus coverage for these systems conferences is incomplete, missing a total of 123 papers relative to GS. For the remaining 1,965 papers, Fig. 8 shows a scatter plot of every paper’s citation count in Scopus (x-axis) and GS (y-axis). The vast majority of points lie above the 1:1 intercept line, confirming that GS citation counts tend to be higher than those of Scopus. But the difference in magnitudes appears to be remarkably constant, with near-perfect correlation (r = 0.99), (p < 10−9), (R2 = 0.98). The implication here is that all the relative observations drawn so far on GS citations should generalize to Scopus citations as well, up to a constant factor. Other studies have found that Google Scholar citations are strongly correlated with those from Web of Science as well (Kousha & Thelwall, 2007).

Figure 8 Comparison of GS and Scopus citations after five years.

Orange line denotes a 1:1 mapping.

RQ4: Effect of self-citations

Self-citations are fairly common in the sciences and have been estimated to comprise 10–40% of all scientific production, depending on the field (Aksnes, 2003; Snyder & Bonzi, 1998; Wolfgang, Bart & Balázs, 2004). On the one hand, self-citations represent a natural evolution of a research team’s work, building upon their previous results, especially in systems projects that often involve incremental efforts of implementation, measurement, and analysis (Wolfgang, Bart & Balázs, 2004). On the other hand, self-citations can be problematic as a bibliometric measure of a work’s impact because they obscure the external reception of the work and are prone to manipulation (Waltman, 2016).

For the final research question, let us quantify the degree to which self-citations affect the overall citation metrics in computer systems and how well self-citations can be predicted from the papers’ references themselves. To this end, we will examine how self-citations evolve over time, how they relate to self-citations in the original papers, and how common they are overall.

Ratio of self-citations over time

We can start by computing the ratio of self-citations for every paper at every measurement time point (the end of each calendar year between 2017 and 2022, which represents different ages in months for each paper depending on the month they were published in). We can then average these ratios across all papers with the same measurement age, and divide those into four quartiles based on the total citations they had at the time. The results of this computation are shown in Fig. 9. A few notable observations emerged from the data:

Figure 9 Mean ratio of self-citations out of total citations as a function of months since publication.

Trend lines are smoothed using LOESS. The data is divided into four quartiles based on how many total citations the paper had after five years (Q4 means most cited papers).

• In the first few months after publication, most of the detected citations appear to be self-citations for all groups. A distinct transition to majority external citations starts about a year after publication.

• Highly cited papers (e.g., Q3 and Q4) are also cited earlier.

• The more a paper is cited overall, the lower the ratio of self-citations. Conversely, for Q1 and Q2, nearly all early citations were self-citations.

• That said, even in the fourth quartile, self-citations still comprise some 10% of all five-year citations on average.

Overall, it appears that self-citations are more characteristic of early citations, either because a paper has not had enough time to be well-known by almost anyone other than its authors, or as an attempt by the authors to increase its visibility (Chakraborty et al., 2015). But as time passes and papers are evaluated by the community, highly cited papers accrue citations primarily from external researchers.

Backward and forward self-citations

These self-citation distributions inspire another curious question: Is there a pattern of specific self-citation behavior that affects both forward and backward citations? In other words, can we predict the number of self-citations a paper would receive after five years by counting self-citations in its own bibliography?

To answer this question, the total number of backward self-citations was estimated, as follows: First, the reference list from each PDF file was converted to a text file for every single paper and manually cleaned up of conversion artifacts such as two-column papers breaking down the reference list. Then, the last names of all authors were searched for in every single reference. Any reference that matched one or more names was counted as a self-citation. Clearly, this method is imperfect because some last names may match different authors, and some last names may even represent a word in a paper’s title. Nevertheless, manual inspection of several dozen papers has revealed very few inaccuracies.

The self-reference counts were converted to ratios by dividing by the total number of references in each paper and then compared to eventual (5-year) self-citations. Figure 10 shows the relationship between these two measures. We can make out several horizontal bands for low-cited papers with fixed ratios of eventual citations (0%, 50%, 100%, etc.), and one clear vertical band for papers with zero backward self-references. But overall, it shows little discernible patterns.

Figure 10 Relationship between five-year self-citations and self-citation count in each paper’s own references.

Orange line denotes a 1:1 mapping.

Likewise, the Pearson correlation between backward and forward self-citation ratios is nominal (r = 0.16), (p < 10−9). Removing some of the outliers (the top and bottom horizontal bands, as well as the leftmost band) does little to increase the correlation (r = 0.19), (p < 10−9). Even if we look only at the temporal subset of data when papers first exceed five citations (which typically comprise of a higher ratio of self-citations), the observed correlation is not particularly strong (r = 0.24), (p < 10−9). It appears that the answer to this research question is mostly negative.

Overall self-citation ratios

One anecdotal observation we can make from Fig. 10 is that most papers appear above the orange 1:1 line. In other words, a majority of papers (1,136 or 57.1% to be exact) had a higher self-citation ratio after five years than their backward self-citation ratio in their reference list.

Moreover, the mean self-citation ratio after five years, 24.1%, is on the one hand higher than the 15% rate found for physical sciences (and 9% over all sciences) in an earlier study on self-citation patterns across disciplines (Snyder & Bonzi, 1998); but on the other hand, is in close agreement with the 24% rate found for CS papers in Norway (Aksnes, 2003). The Norway study also found a high ratio of self-citing papers overall, agreeing with our data where 78.4% of papers had at least one self-citation after five years.

In terms of outliers, hidden in these ratios and averages are several papers whose total citations were predominantly self-citations. In their final citation count, 252 papers had over 50% self-citations, of which 11 papers were in Q4 with dozens of citations each. Four of the six most self-cited papers in relative terms were in architecture conferences, suggesting that research in this area often builds primarily on past research from the same group. That said, the overall correlation between a paper’s total citations and self-citations after five years is low (r = 0.19), (p < 10−9), suggesting that for this dataset as a whole, self-citations are not typically a dominant component of overall citations.

Discussion

This section integrates results from the previous section to explore three topics beyond raw citation counts: the use of H-index as a measure of conference quality, the relationship between a conference’s acceptance rate and its eventual citations, and the characteristics of papers with early citations.

Conference H-index

One derivative aspect of citations that we can calculate is a conference’s H-index. H-index was developed by Hirsch to measure the impact of individual researchers (Hirsch, 2005). It is defined as the maximum integer n such that there exist n papers published in a given time window that received at least n citations. GS also reports the H-index and five-year H-index of conferences and journals and ranks them accordingly (Patience et al., 2017). With the dataset of papers from 2017, we can compute a similar measure based on papers from that single year, if nothing else, to illustrate the weakness of this metric when applied to conferences.

Our partial H5-index metric ranges from 4 for HCW to 69 for CCS. It correlates strongly with the size of the conference (r = 0.75), (p < 10−9), suggesting that a conference’s mere number of accepted papers, largely a policy decision, has a significant impact on its H-index metric. Moreover, recall the observation that the larger a conference, the more likely it is to have a “runaway” successful paper. This suggests that among all the factors that impact a conference’s total citations (and consequently its H-index), there is also a non-negligible element of chance. In other words, a conference’s steering committee could conceivably increase its H-index (and average citations) by simply increasing the size of the conference. Obviously, size is not the most important factor, and an overly permissive acceptance policy is likely to admit papers that would lower the overall citation average. As demonstrated in our dataset, the partial H5-index metric is clearly negatively correlated with acceptance rate (r = −0.62), (p < 10−5). Nevertheless, this finding weakens the case for the use of these metrics as reliable indicators of conference quality.

Conference acceptance rate

The strong correlation between a conference’s acceptance rate and its median citation counts could suggest a causal relationship, that is, that competitive peer review selects for high-impact papers. However, the peer-review process is notoriously unreliable when it comes to selecting highly cited papers, which may not always even be the reviewers’ goal (Coupé, 2013; Lee, 2019; Wainer, Eckmann & Rocha, 2015). Moreover, even if a causal relationship does exist, its direction is unclear. That is to say, researchers who predict that their paper will have a relatively low citation impact may self-select to submit it to a less competitive conference to increase its chances of acceptance. Since we regrettably cannot design a randomized controlled trial where some papers are randomly accepted in a given conference, and since we cannot accurately predict the citation impact of rejected papers, we do not have the necessary tools to evaluate the strength and direction of any such causal link. That said, even if the peer-review process has little predictive or selective power for high-impact papers, we could surmise that successfully publishing in competitive conferences increases the likelihood of eventual citations because intuitively, prestigious conferences increase the post-publication visibility and credibility of their papers.

Early cited papers

Although we can expect several months to pass before a paper is first discovered and cited by external scientists, some “early” papers were cited within six months of publication. Our dataset provides evidence for two possible explanations.

First, many of these citations could be self-citations, which do not require discovery. As we have seen for RQ4, more early citations are indeed self-citations. For example, the 255 “early” papers that were cited within six months of publication (Scopus data) averaged 55.61% self-citations. In contradistinction, in the 463 papers first cited in the following six-month period (months 7–12), this proportion was significantly lower at 41.98% (t = 3.79), (p < 10−3).

Second, the availability of preprints or other freely accessible versions of the paper before publication could accelerate the discovery process. The dataset includes the time (in months) it took GS to discover a freely available e-print version of each paper. For the “early” papers, this time averaged 3.76 months, compared to 5.55 months for the slow papers (t = −3.19), (p < 0.01). For a frame of reference, this average was close to the overall average time to e-print across all papers (5.47 months). From this perspective, therefore, the papers cited within the second half-year are indistinguishable from all later-to-cite papers, as opposed to the distinct “early” papers cited in the first half-year.

Related work

Citation analysis is an active area of research in bibliometrics, with many studies looking at it both quantitatively—examining citation distributions, as this study does—or qualitatively, examining the strengths, weaknesses, and characteristics of citations as a metric of scholarly impact. This study does not aim to debate the merits of citation-based metrics, so the qualitative aspects will not be reviewed here. The interested reader is referred to the recent book “The Science of Science” for an overview of this debate (Wang & Barabási, 2021).

On the quantitative side, several studies exist that examine some of the metrics discussed in this article, as well as others. Tsay & Shu (2011) analyzed citations in one journal and found that the most cited documents were journal articles, followed by books and book chapters, electronic resources, and conference proceedings. As mentioned in the introduction, in computer science, conference papers take precedence to journal articles (Goodrum et al., 2001; Vrettas & Sanderson, 2015).

Citation analyses collect data from different databases, primarily Google Scholar, Scopus, and Web of Science. Other studies have evaluated these databases for their validity and reliability in citation analysis. Several of those concluded that Google Scholar both has higher coverage and more liberal policies for defining what constitutes a citation (Halevi, Moed & Bar-Ilan, 2017), resulting in higher citation counts overall, as was also found here (Kousha & Thelwall, 2007). This difference can make it difficult to argue about absolute numbers of citations since they vary significantly by database. However, as most of these studies also found—and ours agrees—the citation counts from the three databases appear to be strongly correlated (Harzing & Alakangas, 2016; Martin-Martin et al., 2018). The implication is that comparing citations across papers, conferences, and years should produce similar conclusions, regardless of which reputable database is used for citation data.

Numerous studies examined specific aspects of citation analysis, such as the type of citing document or type of cited document. For example, Harter (1996) measured the citation counts and impact factor of early electronic journals to compare them with traditional journals. A related study looked at how often electronic resources are cited in electronic journals (Herring, 2002). Various other studies have found a potential link between the open sharing of research artifacts and increased citations (Frachtenberg, 2022b; Heumüller et al., 2020).

There exist many citation analyses for various disciplines and venues, too numerous to enumerate here. To the best of the author’s knowledge, this study is the first to analyze citation characteristics across the entire field of computer systems research, and the first to cover the four research areas in one analysis. A few particularly relevant examples of similar studies follow, all focused on computer science fields.

In 2001, Broch (2001) analyzed the citations of one prestigious CS conference, SIGIR, over the years 1997–1999. Unlike this study, it covered three years, allowing the discovery of some trends, but it focused narrowly on one conference only (n = 110) and a relatively short post-publication window. It did find, as this study confirms, that papers that are electronically available tend to collect more citations. In another single-conference study, Frachtenberg (2022a) analyzed the citations of papers in the SIGMETRICS’17 conference and found similar citation dynamics to those of the larger systems field in this more generalized study. Similarly focused, but taking a reverse perspective, Lister & Box (2008) looked at the outgoing citations of the SIGCSE’07 conference and examined in detail the venues and venue types of those cited papers.

Some specific subfields of computer systems also received bibliometric analysis that covered more than a single venue. Iqbal et al. (2019a) looked at five decades of the SIGCOMM conferences, and more broadly, at 18 years of four venues focused on computer networks (Iqbal et al., 2019b). The primary investigation of these longitudinal studies was aimed at changes over time, a dimension this cross-sectional study completely omits by limiting the data to a single year. Rahm & Thor (2005) looked at ten years of five primary database venues and looked at most cited papers, venues, authors, institutions, and countries, as well as citation skew and impact factor. In another subfield of computer systems, namely cloud computing, Khan, Arjmandi & Yuvaraj (2022) recently investigated the most-cited papers and the venues they appeared in. Previously, Wang et al. (2016) also tried to identify important papers and research themes in the subfield using tools from social network analysis.

Most of these comparable studies did not go to the same level of detail on the questions of self citations, database selection, and citation dynamics described in this study. Moreover, although these studies share with the present one their attention on computer science (and some even on computer systems), they are all narrowly focused on a single conference or subfield, and do not cover computer systems broadly. However, we can find several studies on the other end, broadening their view either to all of computer science, or even multiple scientific disciplines at once.

For example, Chakraborty et al. (2015) looked at the growth dynamics of citations of some 1.5 million CS papers and identified several distinct patterns. Devarakonda et al. (2020) looked at some 8 million publications from 20 years and classified them into topics based on their citation patterns. Freyne et al. (2010) compared CS journals to conferences and found that conference papers are generally better cited. They also found that GS citations correlate strongly with Web of Science citations (when available), agreeing with our finding when comparing GS to Scopus. Most recently, Saier, Krause & Färber (2023) presented a very large machine-readable dataset of publication metadata (including citation networks) for all arXiv publications in all fields. Although the present work focuses on the data rather than the analysis, and the papers are all preprints rather than peer-reviewed papers as in the other studies, this dataset should prove very useful for large-scale comparisons and bibliometric analyses of entire fields of study.

Conclusion and future work

This work examined, for the first time, the citation characteristics of an important field of computer science, namely computer systems. It found that compared to other fields and disciplines, systems papers are very well cited: the top-cited papers rank the field among the highest scientific disciplines in citation counts, and only a few papers remain uncited after five years. From a practical perspective, these data serve as a grounding point for future citation analyses, either within computer systems or in comparison to other fields.

The overall ratio of five-year self-citations (24.1%) appears to be higher than in other scientific fields but agrees with the 24% rate found for all computer science papers in Norway. Interestingly, most papers exhibit some self-citations, especially within the first few months since publication (when the free availability of e-print versions of a paper also increases its early citation count). Over time, the self-citation ratio remains above 30% for papers with relatively few citations, while for the most cited 25% of papers, this ratio drops below 10%. Practically speaking, this observation suggests that analyzing self-citation statistics should only be performed from a vantage point of several years after publication.

Competitive conferences with a low acceptance rate tend to have higher citations per paper on average. On the other hand, conferences that accept many papers overall (independent of the acceptance ratio) also exhibit more “runaway success” papers that lift their average citation counts, suggesting the possibility of a random factor. It also suggests that the H5-index metric for conferences is particularly prone to manipulation, as increasing the number of accepted papers is correlated with higher H5-index values on average. Whether the citation count comes from Google Scholar or Scopus did not matter much, as both were strongly correlated (Pearson’s r = 0.99), leading to similar findings for all relative comparisons across papers and conferences.

This work can be extended in several directions. Two obvious extensions along the time axis are to continue collecting citation metrics for the same conferences and to look at additional conference years in order to verify the generalization of these results. A more intricate proposal is to move from the “how many”-type questions on citations in this study to the “why”-type questions. The goal of such an investigation would be to identify which among dozens of factors are most closely associated with increased citation counts in the field of computer systems and hypothesize potential explanations. The author plans to follow up with a study that analyzes the myriad factors that could be associated with citation counts: conference-related, paper-related, and author-related.

At a deeper level, the existing data could also help answer questions on the relationships between the text of each paper and its eventual citations. For example, by using topic modeling on the titles, abstracts, or full text of the papers, we could attempt to identify which topics were particularly impactful in 2017, and perhaps compare these findings to more recent systems papers to see how these topics have evolved. Yet another avenue for exploration of this dataset is the analysis of the various networks the data represents: in coauthorship collaborations, in citations, and in affiliations. These networks can be analyzed using tools from social network analysis to quantify the centrality of various papers, authors, or even topics, tying it back to topic modeling.

Supplemental Information

Appendix A. Detailed Conference List

Each conference is described by its initialism, full name, commencement date, size (number of published papers), acceptance rate (if known), and the homepage containing the program.

1. ASPLOS: ACM International Conference on Architectural Support for Programming Languages and Operating Systems, 2017-04-08. 56 papers, acceptance rate: 17.5%. Homepage: https://dl.acm.org/doi/proceedings/10.1145/3037697

2. ATC: USENIX Annual Technical Conference, 2017-07-12. 60 papers, acceptance rate: 21.7%. Homepage: https://www.usenix.org/conference/atc17

3. CCGrid: IEEE/ACM CCGrid, 2017-05-14. 72 papers, acceptance rate: 25.2%. Homepage: https://www.arcos.inf.uc3m.es/wp/ccgrid2017/

4. CCS: ACM Conference on Computer and Communications Security, 2017-10-31. 151 papers, acceptance rate: 18.1%. Homepage: https://www.sigsac.org/ccs/CCS2017/

5. CIDR: The biennial Conference on Innovative Data Systems Research, 2017-01-08. 32 papers, acceptance rate: 41%. Homepage: http://cidrdb.org/cidr2017/

6. CLOUD: IEEE International Conference on Cloud Computing, 2017-06-25. 29 papers, acceptance rate: 26.4%. Homepage: https://ieeexplore.ieee.org/xpl/conhome/8029867/proceeding

7. Cluster: IEEE Cluster Conference, 2017-09-05. 65 papers, acceptance rate: 30%. Homepage: https://cluster17.github.io/

8. CoNEXT: ACM International Conference on Emerging Networking Experiments and Technologies, 2017-12-13. 32 papers, acceptance rate: 18.7%. Homepage: http://conferences2.sigcomm.org/co-next/2017/#!/home

9. EuroPar: International European Conference on Parallel and Distributed Computing, 2017-08-30. 50 papers, acceptance rate: 28.4%. Homepage: http://europar2017.usc.es/

10. EuroSys: The European Conference on Computer Systems, 2017-04-23. 41 papers, acceptance rate: 21.8%. Homepage: https://eurosys2017.github.io/

11. FAST: USENIX Conference on File and Storage Technologies, 2017-02-27. 27 papers, acceptance rate: 23.3%. Homepage: https://www.usenix.org/conference/fast17/

12. HCW: IEEE International Heterogeneity in Computing Workshop, 2017-05-29. 7 papers, acceptance rate: 46.7%. Homepage: https://ieeexplore.ieee.org/xpl/conhome/7964630/proceeding

13. HiPC: IEEE International Conference on High Performance Computing, Data, and Analytics, 2017-12-18. 41 papers, acceptance rate: 22.3%. Homepage: http://hipc.org/

14. HotCloud: USENIX Workshop in Hot Topics in Cloud Computing, 2017-07-10. 19 papers, acceptance rate: 32.8%. Homepage: https://www.usenix.org/conference/hotcloud17

15. HotI: IEEE Annual Symposium on High-Performance Interconnects, 2017-08-28. 13 papers, acceptance rate: 33.3%. Homepage: https://ieeexplore.ieee.org/xpl/conhome/8063637/proceeding

16. HotOS: ACM Workshop on Hot Topics in Operating Systems, 2017-05-07. 29 papers, acceptance rate: 30.9%. Homepage: https://www.sigops.org/hotos/hotos17/

17. HotStorage: USENIX Workshop on Hot Topics in Storage and File Systems, 2017-07-10. 21 papers, acceptance rate: 36.2%. Homepage: https://www.usenix.org/conference/hotstorage17

18. HPCA: The IEEE Symposium on High Performance Computer Architecture, 2017-02-04. 50 papers, acceptance rate: 22.3%. Homepage: http://hpca2017.org

19. HPCC: IEEE International Conference on High Performance Computing and Communications, 2017-12-18. 77 papers, acceptance rate: 43.8%. Homepage: https://www.computer.org/csdl/proceedings/hpcc-smartcity-dss/2017/17D45VtKipS

20. HPDC: ACM International Symposium on High Performance Parallel and Distributed Computing, 2017-06-28. 19 papers, acceptance rate: 19%. Homepage: http://www.hpdc.org/2017/

21. ICAC: IEEE International Conference on Autonomic Computing, 2017-07-18. 14 papers, acceptance rate: 19.2%. Homepage: http://icac2017.ece.ohio-state.edu/

22. ICPE: ACM/SPEC International Conference on Performance Engineering, 2017-04-22. 29 papers, acceptance rate: 34.9%. Homepage: https://icpe2017.spec.org/

23. ICPP: IEEE International Conference on Parallel Processing, 2017-08-14. 60 papers, acceptance rate: 28.6%. Homepage: http://www.icpp-conf.org/2017/index.php

24. IGSC: IEEE International Green and Sustainable Computing Conference, 2017-10-23. 23 papers, acceptance rate: unknown. Homepage: https://web.archive.org/web/20171122154827/http://igsc.eecs.wsu.edu/

25. IISWC: IEEE International Symposium on Workload Characterization, 2017-10-02. 31 papers, acceptance rate: 37.3%. Homepage: http://www.iiswc.org/iiswc2017/index.html

26. IMC: ACM Internet Measurement Conference, 2017-11-01. 28 papers, acceptance rate: 15.6%. Homepage: http://conferences.sigcomm.org/imc/2017/

27. IPDPS: IEEE International Parallel and Distributed Processing Symposium, 2017-05-29. 116 papers, acceptance rate: 22.8%. Homepage: http://www.ipdps.org/ipdps2017/

28. ISC: ISC High Performance, 2017-06-18. 22 papers, acceptance rate: 33.3%. Homepage: http://isc-hpc.com/id-2017.html

29. ISCA: ACM/IEEEE International Symposium on Computer Architecture, 2017-06-24. 54 papers, acceptance rate: 16.8%. Homepage: https://www.iscaconf.org/isca2017/

30. ISPASS: IEEE International Symposium on Performance Analysis of Systems and Software, 2017-04-24. 24 papers, acceptance rate: 29.6%. Homepage: http://www.ispass.org/ispass2017/

31. MASCOTS: IEEE International Symposium on the Modeling, Analysis, and Simulation of Computer and Telecommunication Systems, 2017-09-20. 20 papers, acceptance rate: 23.8%. Homepage: http://mascots2017.cs.ucalgary.ca/

32. MICRO: IEEE/ACM International Symposium on Microarchitecture, 2017-10-16. 61 papers, acceptance rate: 18.7%. Homepage: https://www.microarch.org/micro50/

33. Middleware: The Annual Middleware Conference, 2017-12-11. 20 papers, acceptance rate: 26%. Homepage: https://middleware2017.github.io/middleware2017/

34. MobiCom: ACM International Conference on Mobile Computing and Networking, 2017-10-17. 35 papers, acceptance rate: 18.8%. Homepage: https://sigmobile.org/mobicom/2017/

35. NDSS: The Network and Distributed System Security Symposium, 2017-02-26. 68 papers, acceptance rate: 16.1%. Homepage: https://www.ndss-symposium.org/ndss2017/

36. NSDI: USENIX Symposium on Networked Systems Design and Implementation, 2017-03-27. 42 papers, acceptance rate: 16.5%. Homepage: https://www.usenix.org/conference/nsdi17/

37. PACT: IEEE/ACM International Conference on Parallel Architectures and Compilation Techniques, 2017-09-11. 25 papers, acceptance rate: 23.1%. Homepage: https://www.computer.org/csdl/proceedings/pact/2017/12OmNwbcJ4M

38. PODC: ACM Symposium on Principles of Distributed Computing, 2017-07-25. 38 papers, acceptance rate: 24.7%. Homepage: https://www.podc.org/podc2017/

39. PODS: ACM Symposium on Principles of Database Systems, 2017-05-14. 29 papers, acceptance rate: 28.7%. Homepage: http://sigmod2017.org/pods-program/

40. PPoPP: ACM SIGPLAN Symposium on Principles and Practice of Parallel Programming, 2017-02-04. 29 papers, acceptance rate: 22%. Homepage: http://ppopp17.sigplan.org/

41. SC: The International Conference for High Performance Computing, Networking, Storage and Analysis, 2017-11-13. 61 papers, acceptance rate: 18.7%. Homepage: http://sc17.supercomputing.org/

42. SIGCOMM: ACM SIGCOMM Conference, 2017-08-21. 36 papers, acceptance rate: 14.4%. Homepage: http://conferences.sigcomm.org/sigcomm/2017/

43. SIGMETRICS: ACM SIGMETRICS, 2017-06-05. 27 papers, acceptance rate: 13.3%. Homepage: http://www.sigmetrics.org/sigmetrics2017

44. SIGMOD: ACM International Conference on Management of Data, 2017-05-14. 96 papers, acceptance rate: 19.6%. Homepage: http://sigmod2017.org/

45. SOCC: ACM Symposium on Cloud Computing, 2017-09-25. 45 papers, acceptance rate: unknown. Homepage: https://acmsocc.github.io/2017/

46. SOSP: Symposium on Operating Systems Principles, 2017-10-29. 39 papers, acceptance rate: 16.8%. Homepage: https://www.sigops.org/sosp/sosp17/

47. SP: IEEE Security and Privacy, 2017-05-22. 60 papers, acceptance rate: 14.3%. Homepage: https://www.ieee-security.org/TC/SP2017/index.html

48. SPAA: ACM Symposium on Parallelism in Algoirmths and Architectures, 2017-07-24. 31 papers, acceptance rate: 24.4%. Homepage: http://spaa.acm.org/2017/index.html

49. SYSTOR: ACM International Systems and Storage Conference, 2017-05-22. 16 papers, acceptance rate: 34%. Homepage: https://www.systor.org/2017/

50. VEE: ACM International Conference on Virtual Execution Environments, 2017-04-09. 18 papers, acceptance rate: 41.9%. Homepage: http://conf.researchr.org/home/vee-2017

Additional Information and Declarations

Competing Interests

Author Contributions

Data Availability

The author declares that he has no competing interests.

Eitan Frachtenberg conceived and designed the experiments, performed the experiments, analyzed the data, performed the computation work, prepared figures and/or tables, authored or reviewed drafts of the article, and approved the final draft.

The following information was supplied regarding data availability:

The data is available at Zenodo:

Eitan Frachtenberg. (2021). eitanf/sysconf: artifact-revised (artifact-revised). Zenodo. https://doi.org/10.5281/zenodo.5590575

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
