# Peer review of "Citation analysis of computer systems papers"

_PeerJ Computer Science, doi:10.7717/peerj-cs.1389_

## Round 0.1 · original submission · Minor Revisions

Dear Author,

Thank you for submitting your manuscript to PeerJ Computer Science for consideration. We appreciate the time and effort you have invested in your work. After receiving feedback from two reviewers who thoroughly evaluated the manuscript, we have determined that minor revisions are necessary before your manuscript can be accepted for publication.

Both reviewers have provided constructive feedback and suggestions for improvement. We kindly ask you to address the concerns raised by the reviewers in your revised manuscript. Please find the specific comments from each reviewer below. Both reviewers have put forth several suggestions that could significantly enhance your work. There are various aspects that need clarification and elaboration, such as the novelty, contributions, and findings, as well as their potential implications. Additionally, improvements should be made to the readability of certain texts and figures. Another fascinating aspect to explore is the comparison of your work with other fields beyond computer science.

To submit your revised manuscript, please follow the submission guidelines and provide a point-by-point response to the reviewers' comments, detailing the changes made in the manuscript. We look forward to receiving your revised manuscript and considering it for publication.

Thank you for considering our journal as a venue for your research. We appreciate your contribution to the scientific community and eagerly await your revised manuscript.

Sincerely,

José Manuel Galán

·

Basic reporting

I have thoroughly reviewed the manuscript and would like to provide comprehensive feedback and suggestions to enhance the quality of the paper.

Abstract:

1. Instead of mentioning "four research questions," consider discussing "four research topics" or "four areas of investigation." This would establish a better connection between the abstract and the actual content of the paper.
2. Connect the findings such as "the most cited subfields" to the "overall distribution of systems citations" as part of the findings within a specific topic.
3. Focus on one or three main contributions and their practical implications to make the paper more focused and easier to understand. The other findings could still be discussed but giving priority to the main contributions would make the abstract more cohesive and the overall paper more compelling.

Introduction:

1. Provide examples or specific use-cases to emphasize the importance and relevance of citation analysis in the context of computer systems.
2. Expand on the purpose of the paper and how it contributes to the existing body of knowledge in citation analysis.
3. Emphasize the novelty and relevance of the research by providing more context, discussing the gaps in existing research, and highlighting how the study's methodology or findings address those gaps.
4. Remove the subtitles in the introduction and incorporate the content as part of the main text.
Present a high-level overview of the research questions and their potential contributions, rather than delving into the detailed answers at this stage.
5. Expand the contributions section to provide more insight into the paper's main contributions and their practical implications.

Experimental design

Materials and Methods:

1. Provide a flowchart showing the data collection process and a more detailed explanation of the methodology.
2, Clarify how the dataset was obtained and processed, either within the main text or in a separate paper.
3, Present the distribution of citations in the form of descriptive statistics within the text instead of Figure 1.
4. Provide a clearer explanation of the citation tracking methodology.

Validity of the findings

Results:

1. Utilize topic modeling methods to identify the topics of the conferences using the titles and abstracts of the papers. (suggestion or for future research)
2. Check line 257 for clarity and accuracy.
3. Consider merging figures 4, 5, and 6 to make it easier for readers to compare the information presented in these figures.
4. In the conclusion section, emphasize the practical implications of the study.
5. Explore collaboration networks in future research using the available dataset.

Additional comments

The authors possess a valuable dataset that is specifically focused on computer science, making the topic highly relevant and well-suited for this journal.

Cite this review as

Reviewer 2 ·

Basic reporting

The paper is very easy to read and clearly structured.

It is also very valuable that the authors provide the data (lines 83-84) and also the code (lines 122-123), which increases the reproducibility of the paper.

Citations and related work are extensively curated to the best of my knowledge. However, I would like to inform the authors about a new, very relevant paper on citation analysis [Saier et al., 2023]. Saier et al. (2023) create a **public** database of 1.9 million publications from the arXiv repository, spanning multiple disciplines and 32 years. They also do some preliminary analysis (not comparable to this paper because of the different scope). It is a preprint, but the amount of data and analysis they do is worth reading. Both papers are concurrent, so this is not a missing citation, but a suggested paper to keep in mind.

Regarding Figure 7, I have trouble seeing the citation velocity of the papers. From my point of view, the amount of semi-transparent lines inside the small box for each conference makes it very difficult to understand. I might make a sense of the overall trend (the more dark bottoms, the more citation velocity?). But I'd suggest a different way to do it, e.g. an average for all papers, so you only have one line for each conference.

Saier, T., Krause, J., & Färber, M. (2023). unarXive 2022: All arXiv Publications Pre-Processed for NLP, Including Structured Full-Text and Citation Network. arXiv preprint arXiv:2303.14957.

Experimental design

The author's experimental design seems aligned to answer the proposed questions. However, I miss a comparison, advantages and downsized with the experimental design that other works about blibliometric analysis in CS did. The author already cites these works in the intro, but I miss the comparison. Does the author add any new analysis? I suggest just adding a few sentences about this comparison [Broch, 2001, Frachtenberg, 2022b, Iqbal et al., 2019a,b, Lister and Box, 2008, Rahm and Thor, 2005, Wang et al., 40 2016].

In addition, the authors state that they compare the distribution of citations, the evolution of citations, etc. between different fields (e.g. line 48). But then they focus only on computer systems. Although I can see the point and the clear contribution to the field of computer systems, I think that the statement about the comparison with other fields in computer science is a bit exaggerated for the experiments the author does. In my opinion, just focusing on the computer systems subfield is a contribution significant enough to be worthy. Therefore, I could suggest 2 things: either the author lower the statement about the comparison with other fields in CS, or they perform a bigger analysis of the comparison between computer systems and other subfields.

I agree that limiting citations is not the best way to make an impact (pp. 141-143). However, I think it is the best current metric (considering the effort-utility tradeoff) that we can use as a proxy for publication impact. So I agree with the authors that that choice does not lower the significance of the work.

Validity of the findings

I really like the way the authors present the results. First, they present the specific quantitative findings along with the research questions in the introduction of the paper (51-77). Then they list the qualitative - or high level - findings in lines 78-86, which makes it easy to understand the paper.
The research questions and findings are relevant. However, they only analyze papers published in 2017, which limits the robustness of the findings. Although they use a large number of papers, the dynamics of this year may change in other years. Therefore, the analysis would benefit from a broader range of publication years. Instead, if the authors insist on using only one year, they should refer to it more often in the manuscript so that the conclusions are not misinterpreted.

Given the current state of CS academia, I think the finding that Google Scholar citations, while higher, are still comparable to other sources in paper-to-paper comparisons is very valuable for our field. As noted above, a broader analysis in fields and publication years will strengthen this finding, although I agree that this is beyond the scope of the paper.

In line with my previous concerns about the comparison with other fields. The authors give a lot of data analysis on the computer systems field, but nothing on the other fields. This should not be a problem when analyzing the computer systems field. However, they have statements like line 171 where they clearly refer to the other fields without providing data for the other fields. My suggestion is the same as before: the work would benefit either if you added that data for a clear comparison, or if the sentence focused only on the computer systems fields, avoiding comparison with other fields.

Cite this review as
Anonymous Reviewer (2023) Peer Review #2 of "Citation analysis of computer systems papers (v0.1)". PeerJ Computer Science

---

## Round 0.2 · accepted · Accept

I am pleased to inform you that your manuscript has been accepted for publication in PeerJ Computer Science.

I want to commend you for the significant improvements you have made to your manuscript in response to the comments and suggestions of the reviewers.

Your revisions have resulted in a more substantial and compelling paper that we believe will significantly interest our readers.

We want to thank you for submitting your work to our journal, and we look forward to sharing your findings with the scientific community.

Once again, congratulations on your successful research, and we wish you continued success in your research endeavours.

Sincerely,

José M. Galán